# Deep Learning for Rapid Identification of Microbes Using Metabolomics Profiles

**DOI:** 10.3390/metabo11120863

**Published:** 2021-12-13

**Authors:** Danhui Wang, Peyton Greenwood, Matthias S. Klein

**Affiliations:** 1Department of Food Science and Technology, The Ohio State University, Columbus, OH 43210, USA; dw554@cornell.edu (D.W.); greenwood.121@buckeyemail.osu.edu (P.G.); 2Department of Food Science, University of Massachusetts, Amherst, MA 01003, USA

**Keywords:** artificial neural networks, machine learning, food safety, NMR, pathogens

## Abstract

Rapid detection of viable microbes remains a challenge in fields such as microbial food safety. We here present the application of deep learning algorithms to the rapid detection of pathogenic and non-pathogenic microbes using metabolomics data. Microbes were incubated for 4 h in a protein-free defined medium, followed by 1D ^1^H nuclear magnetic resonance (NMR) spectroscopy measurements. NMR spectra were analyzed by spectral binning in an untargeted metabolomics approach. We trained multilayer (“deep”) artificial neural networks (ANN) on the data and used the resulting models to predict spectra of unknown microbes. ANN predicted unknown microbes in this laboratory setting with an average accuracy of 99.2% when using a simple feature selection method. We also describe learning behavior of the employed ANN and the optimization strategies that worked well with these networks for our datasets. Performance was compared to other current data analysis methods, and ANN consistently scored higher than random forest models and support vector machines, highlighting the potential of deep learning in metabolomics data analysis.

## 1. Introduction

About 10% of the world’s population is infected by foodborne disease per year, causing economic loss, hospitalizations, and death [1]. Efforts to curb the spread of pathogens through food products include strict sanitary protocols, new technologies for inactivating food microbes, and testing for microbial contaminations [2]. Despite these efforts, outbreaks of foodborne disease remain a common occurrence. One reason for these outbreaks is that food microbial testing can be a slow process. Current methods for the detection of food microbes include classical microbiological methods such as plating and qPCR analysis. In some cases, tests results are available after periods of 24 h or more, which may be too slow to stop the distribution of contaminated food products. Metabolomics, the comprehensive measurement of metabolites in a system, is a promising technique in this case because of its speed in delivering results [3]. Nuclear magnetic resonance (NMR) metabolomics has been proposed as a method to distinguish between *E. coli* and *Shigella* strains [4].

Artificial neural networks (ANN) are a type of computational algorithm inspired by the layered structure of the brain’s neurons and its ability to learn from, and classify, visual information [5]. ANN usually consist of layers of mathematical functions called “neurons” in reference to the nerve cells of the same name. Each neuron can take multiple inputs and deliver a single numeric output value. For this, each numeric input value is multiplied by a factor or “weight”, where high weights cause an input to have a large influence on the output, while low weights mean that this input has little or no effect on the output. The output value is calculated by applying a so-called activation function on the sum of inputs multiplied by their respective weights.

An example for a multiplayer ANN is shown in Figure 1. In this example, numeric data of four features are used as input data. ANN have a first layer of “input neurons”; these neurons can either pass through the original value or perform data scaling or transformation. Next, one or multiple “hidden layers” of neurons are added. In a “dense” (fully connected) layer, all neurons receive input values from all neurons in the previous layer. Finally, there is an output layer of neurons. In a classification problem, each output neuron might be associated to one possible output value; in our example in Figure 1 these outputs have two possible values: “*Salmonella*” and “Not *Salmonella*”. The output neuron with the highest numeric values represents the prediction outcome; in Figure 1, this outcome is *Salmonella*.

After creating the network structure, the network needs to be trained on training data. In this step, the weights of the individual neurons are adapted to gain the desired result by means of backpropagation. One “epoch” of training is defined as a training cycle in which the network is trained once on each sample of the training dataset. To achieve a well-trained ANN, multiple epochs of training may be required. After training or learning, the network can be used to predict unknown data inputs. The quality of the outputs of an ANN can be either assessed by calculating accuracy, or by calculating the “loss”, a measure of the difference of the predicted output values to the expected (true) output values. The goal of training or learning is minimizing loss and maximizing accuracy.

Multiplayer ANN, as shown in Figure 1, are commonly known as deep neural networks. Deep learning algorithms have been suggested as a powerful tool in the metabolomics field but have so far not been widely employed [6,7].

The aim of the presented study is to test the potential of metabolomics techniques combined with deep learning algorithms to rapidly detect pathogenic and non-pathogenic microbes.

## 2. Results

The dataset consisted of NMR spectra from 10 different microbial strains, resulting in 80 spectra. Spectra were binned with a bin width of 0.005 ppm. After noise removal and removal of the solvent area, each sample had 1384 bins (features). To account for slight differences between batches of medium, a binned spectrum of pure medium was subtracted from each binned spectrum after microbial growth.

### 2.1. Unsupervised Approach

A principal component analysis (PCA) plot revealed strong differences between microbes (Figure 2). Partial group separations are visible for most stains. *Salmonella* was the only strain that showed complete separation from all other samples. The two *E. coli* strains clustered together and partially overlapped. The loadings plot of this PCA (Appendix A) shows that acetic acid contributed strongly to the observed group separation.

### 2.2. ANN Training

To predict the microbial strain based on metabolite profiles, we created and trained ANN models. Initial tests showed that using two hidden layers increased accuracy by 3.8% as compared to ANN with one hidden layer, while using three or more hidden layers did not further increase accuracy. Using hidden layers of equal size of 800 neurons each yielded consistently good accuracies. Tests of different activation functions showed that rectified linear unit (ReLU) [8] yielded better results for these data than Leaky ReLU, linear, sigmoid, and tanh functions. During training, the optimizer Adam (adaptive moment estimation) [9] yielded 7.7% higher accuracies than stochastic gradient descent (sgd). Therefore, we employed ANN with 2 × 800 hidden neurons, ReLU activation, and Adam for optimization in all further analyses.

Figure 3 shows the training behavior of these models on our dataset. It is interesting to note that the accuracy of the internal cross-validation (solid green lines) reached 100% after less than 30 epochs of training; however, this finding was not accompanied by maximum accuracy in the outer cross-validation (dashed red line). It is, thus, of importance to use outer cross-validation to estimate true accuracies. During further training, internal accuracy stayed at 100%, but loss decreased continuously, and external accuracy increased, until about 300 epochs of training. Then, a steep increase in loss and decrease in accuracy occurred. After further training, better results were obtained until epoch 550, when further accuracy decreases started occurring. At these points, the models get worse when continuing learning through the effect of overfitting. This means that the model is learning the outcomes of the given training samples “by heart” without being able to generalize this knowledge. To avoid overfitting, we need to define ways to stop training before accuracy decreases to find optimal models.

Based on these observed training behaviors, we created a set of criteria to select the correct amount of learning (Figure 4):

Case (a): If the loss function decreases after an epoch of training, training continues. This scenario is also known as a “greedy” algorithm.

Case (b): Training may continue even if the loss function increases, but only if the new loss value is lower than the maximum loss value from the previous seven epochs of training. Visually, this means the algorithm can “jump” out of narrow and shallow valleys. Without this criterion, training might end when reaching a local minimum, but missing the global minimum.

Case (c): Training stops if loss increases more than the maximum value of the past seven epochs. This criterion means the algorithm stops in broad and/or steep valleys that might indicate a global minimum.

Case (d): Training always stops if the new loss value exceeds 2.25 times the lowest loss observed during training, even if the value is lower than the maximum of the last seven epochs. This number was empirically chosen based on analysis of initial tests, where such a steep increase in loss was usually observed shortly before a huge increase in the loss function.

In rare cases where a loss value of exactly zero is observed, this value is replaced by a very low value of 2 × 10^9^ to allow for the calculations for the above criteria. This value was chosen as it was in the range of the lowest non-zero loss values observed in our data.

The criteria defined above were then used for training the actual ANN models to estimate prediction accuracy by outer cross-validation.

Table 1 shows the prediction accuracies when using all signals in the dataset to create ANN models. ANN models were able to correctly predict 91.2% of unknown microbes in this case. We expected some of the metabolites produced during microbial growth to be specific for the respective microbial strain. Therefore, in a second step, we repeated the training of classification models, this time using only metabolite signals that increase during the incubation time. ANN now reached 99.2% accuracy (Table 1).

### 2.3. Method Comparison

To compare the results of the ANN in the realm of more commonly used metabolomics methods, we conducted random forests (RF) and linear support vector machines (SVM) in an identical cross-validation setup. Table 1 shows the prediction accuracies when using all signals in the dataset to create the models. RF and SVM showed lower results than ANN. Using only increasing signals, accuracy increased for all tested models.

Table 2 shows final prediction accuracies of ANN per microbial strain when using only increasing signals. Most strains were predicted correctly in all cases. Interestingly, the two *E. coli* strains were correctly identified in all cases, despite their partial overlap in the PCA.

To assess the metabolic differences between strains, *t*-test were performed to identify signals that changed significantly during growth. *P*-values were corrected using the FDR method. Figure 5 shows volcano plots to visualize the significant signals for each microbial strain, including the metabolites that could be assigned to significant signals by database searches. Metabolite identities and respective chemical shift values can be found in Appendix A.

Table 3 shows a list of identified metabolites that were significantly increasing in a specific microbe. Some metabolites, such as acetic acid, increased in most of the analyzed microbes, while others were more specific and found in only a few of the microbial samples under the given growth conditions.

## 3. Discussion

In this study, we cultivated various common microbes in order to analyze differences in their metabolic products. A future application of this research could be the identification of unknown microbes. In these cases, one cannot select a growth medium that is optimal for growth of the respective microbe as the microbial strain will be initially unknown. Therefore, we needed to choose a medium that allows for growth of various microbial strains. The molecular composition of the medium will influence the growth behavior and the profile of secreted microbial metabolites. To achieve reproducible results, the chemical composition of the medium needs to be well-defined. Rich media such as Muller Hinton medium are based on extracts of animal tissues and may thus show large batch differences in their molecular make-up based on the used starting material. These differences can impede long-term reproducibility. Therefore, we chose a new, protein-free, chemically defined medium that allows cultivation of a broad range of microbes for this study [10]. Media was inoculated with 10^6^ CFU/mL of each microbial strain and medium samples were collected after 4 h of growth. 1D ^1^H NMR spectra were recorded for all samples and analyzed using an untargeted binning approach.

ANN with two hidden layers of 800 neurons each were created and trained according to an optimized set of training criteria, avoiding shallow local minima but stopping at unusually steep increases in the loss function (Figure 4). These training criteria resulted in excellent predictive results for unknown samples (Table 2). Simple feature selection, in this case selecting only increasing signals, was shown to improve the accuracy of the ANN models. It is of importance that this feature selection step must happen within the cross-validation loop (after removal of test samples), and may not happen outside of this loop, as this would use information from the test samples during training and invalidate the results. Overall, ANN allowed for very accurate prediction of unknown microbial samples grown under lab conditions. Especially, ANN accuracies were higher than those of two other common machine learning algorithms, namely RF and SVM. RF had slightly higher accuracies than SVM in both datasets (all signals/only increasing signals), which is in line with previous reports [11].

To further investigate metabolic differences between the analyzed microbes, we performed additional untargeted analyses. PCA analysis revealed partial or complete group separation for most microbial strains (Figure 2). Loadings plot analysis revealed that acetic acid was strongly contributing to this separation, indicating it is present at higher levels after growth of microbes such as *Salmonella* and *E. coli* (Appendix A). Statistical analysis indeed revealed significantly elevated levels of acetate for all analyzed microbes except for *Pseudomonas* (Figure 5). A closer analysis of the NMR signal of acetate (Figure 6) revealed that acetate is visibly produced during growth of *Pseudomonas* when compared to a medium sample, but at much lower rates than in microbes such as *Salmonella*. This small release of acetate did not reach the significance level in the presented study though. Aerobic production of acetate has been reported for *Bacillus* [12], *Candida* [13], *E. coli* [14], *Listeria* [15], *Salmonella* [16], *Shigella* [17], *Staphylococcus* [18], and *Yersinia* [19].

Ethanol was significant in *E. coli* O157:H7 and *Salmonella*. Ethanol is a known microbial metabolite that is mainly associated with anaerobic growth of *E. coli*, but it has been shown that *E. coli* can produce significant amounts of ethanol under aerobic conditions as well [20]. Similarly, ethanol secretion in *Salmonella* is mostly known from anaerobic growth. Still, aerobic production of ethanol has been previously reported for *Salmonella* and it was hypothesized that intracellular microcompartments allow for this effect [16].

1-Propanol is produced under aerobic conditions in *E. coli* [20], *Salmonella* [21], and *Yersinia*. In *Listeria*, 1-propanol has been shown to be produced [22], but not under glucose-rich conditions as in our study.

Formic acid was found elevated in accordance with previous studies on aerobic growth of *Salmonella* [23], but no formate production has previously been reported aerobically in *Staphylococcus* [24]. However, the latter study used a markedly different medium, which could affect the produced metabolites.

Fumaric acid was found to be significantly increased after growth of most of the microbes analyzed in this study. It has been shown that fumaric acid is being secreted during aerobic growth of *E. coli* to facilitate the uptake of L-aspartate via the *DcuA* transporter [25]. L-aspartate is part of the defined medium, making this a possible explanation for the observed increase in fumaric acid.

Indole was secreted in significant amounts in *E. coli* K12, *Staphylococcus*, and *Yersinia* (Table 2). Indole has multiple physiological roles, including intercellular signaling, and is produced by various bacteria [26]. Indole is a metabolite of tryptophan, which is part of the defined medium, thus explaining the observed indole production.

Lactic acid was significantly increasing in *Bacillus* under the employed growth conditions. *Bacillus* has been previously shown to secrete lactic acid in aerobic, glucose-rich conditions [27].

Spermidine was found in the growth medium of both *E. coli* strains, the closely related *Shigella*, *Salmonella*, and *Yersinia*. It has been shown that various common microbes can produces spermidine from amino acids and secrete it into the medium [28]. The employed medium contains a large number of amino acids, which can serve as precursors for the observed spermidine.

Succinic acid was found in several microbes, and a potential pathway for the synthesis of succinic acid from glutamate has been described for *Listeria* [29]. Glutamate is a compound found in the defined medium we used for this study, which would explain the observed succinic acid.

Overall, we found multiple microbial metabolites to be significantly elevated after microbial growth. While some of these metabolites were elevated in many microbes, the individual levels and the combination with more rarely observed metabolites generates a kind of metabolic fingerprint that the ANN can employ to classify different microbes effectively.

Future applications of this technique include the identification of unknown microbes in real-life samples. While we did not attempt this in the presented study, we expect this application to add new analytical challenges that need to be addressed properly before the method will be able to deliver correct predictions. First, real-world samples will usually contain a combination of microbial species rather than a single microbial strain, which will result in significant changes to the obtained metabolite concentrations. This might require adjustments to the experimental protocols, such as optimal growth times, as well as adjustments to the employed data analysis methods. ANN would need to be trained with data stemming from microbial co-cultures to enable the networks to capture this added complexity. In addition, real-life sample collection will introduce contaminations to the growth medium that could hamper correct prediction. We believe the approach we used in this study (subtracting a spectrum of original medium from the final sample) will be beneficial in these cases as well. In addition, proper control and standardization of sample collection and preparation will be required to limit the effects of such matrix effects. Lastly, differences in initial microbial cell counts need to be quantified. This could be achieved by using general linear models based on a targeted set of metabolites after initial microbe identification using ANN.

## 4. Materials and Methods

### 4.1. Microorganism Cultivation

All chemicals were of analytical grade and purchased from Fisher Scientific (Hampton, NH, USA) unless stated otherwise. *Escherichia coli (E. coli)* K12, *E. coli* O157:H7, *Salmonella enterica* serovar Typhimurium, *Listeria innocua*, *Pseudomonas fluorescens*, *Staphylococcus epidermidis*, *Yersinia enterocolitica*, *Bacillus subtilis*, *Shigella flexneri*, and *Candida albicans* were obtained from Professor Ahmed Yousef at the Department of Food Science and Technology of The Ohio State University and stored at –80 °C. Cultures were streaked on tryptic soy agar (TSA) plates and incubated at 30 °C (*P. fluorescens*, *B. subtilis*, and *C. albicans*) or 37 °C (the other strains) for 24 h, respectively. A single colony from each plate was picked, inoculated into 10 mL of tryptic soy broth (TSB) (BD Bacto, Franklin lakes, NJ, USA), and incubated at 30 °C or 37 °C for 20 h with 280 rpm agitation.

Cells were harvested, centrifuged at 7600× *g* for 3 min, washed twice with 0.01 M phosphate buffered saline (PBS, pH 7.4), and resuspended to the original volume in PBS. Optical density at a wavelength of 600 nm (OD_600_) of the suspension was measured using a Biowave Cell Density Meter CO8000 (Biochrom, Holliston, MA, USA) to determine cell density. Then, the cell density of the suspension was adjusted to 10^8^ CFU/mL by the addition of 0.01 M PBS. The concentration of each strain in the suspension was confirmed by plate counting. An aliquot of 100 μL of the suspension was used to inoculate 10 mL of a protein-free defined medium in a 50 mL culture flask to result in a final starting cell count of 10^6^ CFU/mL. The protein-free defined medium was prepared as described recently [10]. Three replicates of each strain were incubated at 37 °C for 4 h, after which medium samples were collected and immediately filtered using a 0.22 µm filter membrane to remove microbes. As a reference, medium samples were also collected immediately following inoculation (time point 0). This procedure was repeated three times with new batches of medium, representing biological replicates.

### 4.2. NMR Measurements

Metabolic profiles were analyzed using Nuclear Magnetic Resonance (NMR) spectroscopy. Six hundred µL of the medium were mixed with 50 µL of deuterium oxide with 0.05% trimethylsilyl-2,2,3,3-tetradeuteropropionate (TSP) as an internal standard and boric acid to prevent bacterial growth. The pH was adjusted to 7.4 before the solution was transferred into 5 mm NMR tubes. 1D ^1^H NOESY spectra were collected at 298 K on an 850 MHz Avance III HD Ascend spectrometer equipped with a cryoprobe (Bruker BioSpin, Billerica, MA, USA). For each sample, the probe was automatically locked, tuned, matched, and shimmed. Spectra were processed in Topspin 3.6.1 (Bruker). For selected samples, ^1^H-^13^C heteronuclear single quantum coherence (HSQC) NMR were measured to aid in metabolite identification.

### 4.3. Data Preparation

An untargeted metabolomics approach was chosen for this study, allowing for a broad analysis of unexpected metabolites, without restricting the dataset to a set of expected metabolites. For this, spectra were binned with a bin width of 0.005 ppm using mrbin (1.6.1) [30]. Spectra were scaled to the internal concentration standard (TSP), but no further scaling or transformation of the signals (e.g., to total spectrum intensity or microbial cell count) was used. Noise bins were automatically removed from the dataset. All statistical analysis were performed in R (3.5.1) (The R Foundation for Statistical Computing, Vienna, Austria). To take into account batch differences of the defined medium, we subtracted a binned spectrum of medium at time point 0 from each binned spectrum. Principal component analysis (PCA) was calculated and plotted to visualize the data.

### 4.4. Classification

To predict microbial identifies, a supervised approach was employed. Artificial Neural Networks (ANN) were created using R packages keras (2.6.0) and tensorflow (2.6.0). The network architecture used two dense, completely linked hidden layers of 800 neurons each with ReLU as activation function. The output layer had ten neurons, one for each analyzed microbial strain, and used a softmax activation function. During ANN training, accuracy was optimized using the Adam optimizer. Models were trained until no further decrease in the loss function (sparse categorical crossentropy) was observed. RF and linear SVM models were created using the R packages caret (6.0–86), randomForest (4.6–14), and e1071 (1.7–6).

For each method (ANN, RF, and SVM), a series of models were generated with an external (outer) leave-one-out (LOO) cross-validation. In this approach, one spectrum is removed from the dataset, then a model is trained, and then the missing sample is predicted using the generated model. This process was repeated until all spectra were predicted once. This approach allows us to estimate the true error of prediction. Accuracy was calculated by taking the mean of the correct estimates per strain divided by the total number of spectra from the respective strain. To analyze the influence of the random numbers used during modeling, this process was repeated six times with different starting seeds for the random number generator.

In a second analysis, the calculations were repeated, this time using only signals that increased during microbial growth compared to a time point 0 medium sample. Increasing signals were picked within the cross-validation loop by selecting signals that increased in all samples containing this strain. This step was performed to put an emphasis on metabolites produced by microbes rather than metabolites consumed, which we expect to be less specific.

### 4.5. Statistical Data Analysis

Two-tailed, unpaired one-sample *t*-tests were calculated for each microbial strain, testing differences of the feature intensity from 0. These tests were repeated for each feature (bin) in the dataset and resulting *p*-values were corrected for multiple testing using FDR controlling at a level of 20% [31]. Volcano plots were used to plot the negative decadic logarithm of the raw *p*-values versus the feature intensity after subtraction of a pure medium spectrum as detailed above. Fold changes were not employed in this version of volcano plots as many features of interest were absent in the pure medium, and fold changes are not defined for starting concentrations of 0. Features that were significantly increasing were identified using database searches in the Human Metabolome Database (http://hmdb.ca/, accessed on 20 November 2021) and the Biological Magnetic Resonance Data Bank (BMRB, http://bmrb.wisc.edu/, accessed on 20 November 2021) and measurements of pure compounds.

## 5. Conclusions

Here, we present a study on the detection of pathogenic and non-pathogenic microbes using metabolomics and deep artificial neural networks. Various microbes were grown in different flasks of identical defined, protein-free medium for 4 h, and medium metabolites were then measured by 1D ^1^H NMR spectroscopy. Artificial neural networks with two hidden layers of 800 neurons each were trained on the data and used to predict unknown samples in an outer cross-validation scheme. Estimation of true error rates showed excellent accuracies, exceeding the results of other current methods such as RF and SVM. Results were highest when using meaningful feature selection, in this case using only metabolite signals that increased during microbial growth (microbial metabolites). To the best of our knowledge, this study is the first to employ deep learning to classify NMR metabolomics data for microbe detection. The results show the potential of these powerful algorithms for metabolomics studies, which will need to be further developed to meet real-world requirements.

## Figures and Tables

**Figure 1 metabolites-11-00863-f001:**
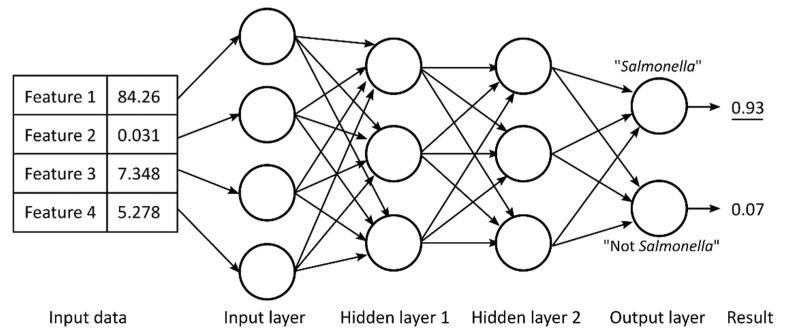
Example of a simple multilayer artificial neural network (ANN) with four input neurons, two fully connected hidden layers of three neurons each, and two output neurons. In this example, the output neuron “*Salmonella*” has the maximum output value, and thus, the ANN predicts that the input data came from a *Salmonella* sample.

**Figure 2 metabolites-11-00863-f002:**
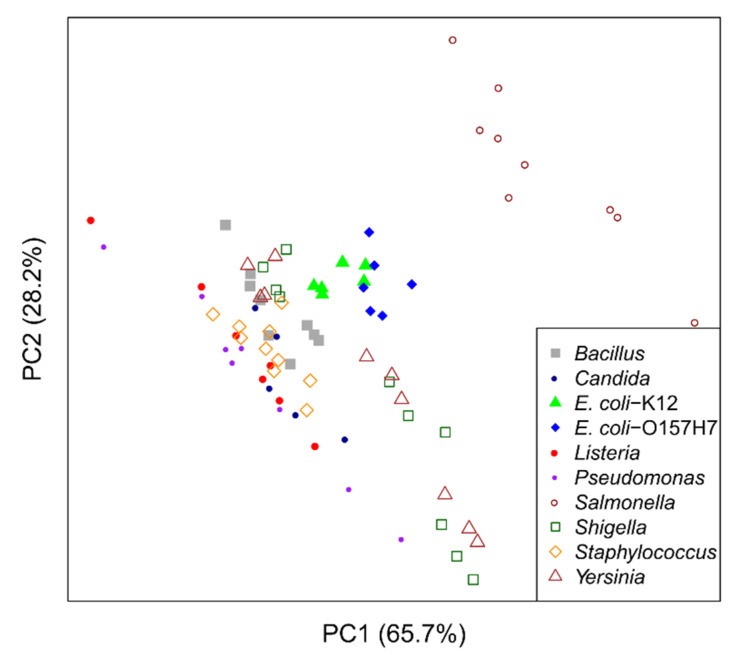
Principal component analysis (PCA) of the 1D ^1^H NMR data. Complete group separation is visible for *Salmonella*, while the other microbial strains show partial group separation.

**Figure 3 metabolites-11-00863-f003:**
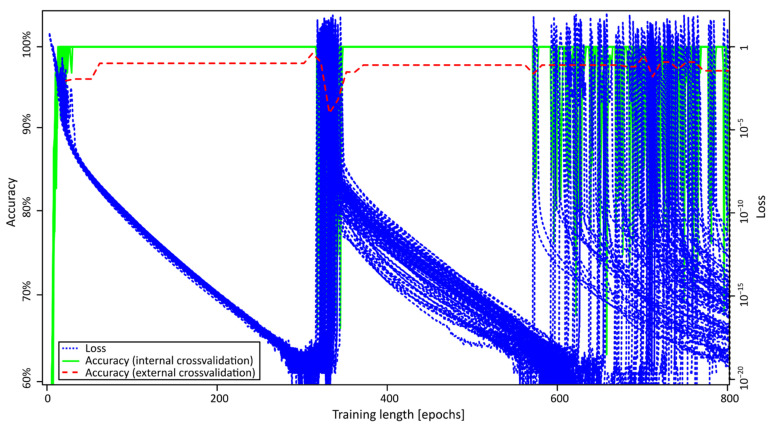
Training behavior of the Artificial Neural Networks. Loss (blue, dotted), accuracy from internal cross-validation (green, solid), and accuracy from external cross-validation (red, dashed).

**Figure 4 metabolites-11-00863-f004:**
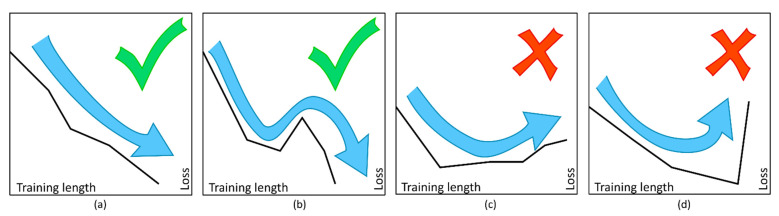
Criteria for training. (**a**) “greedy” learning; (**b**) “jumping” out of narrow and shallow local minima; (**c**) stop at broader minima; (**d**) stop at steep increases in loss.

**Figure 5 metabolites-11-00863-f005:**
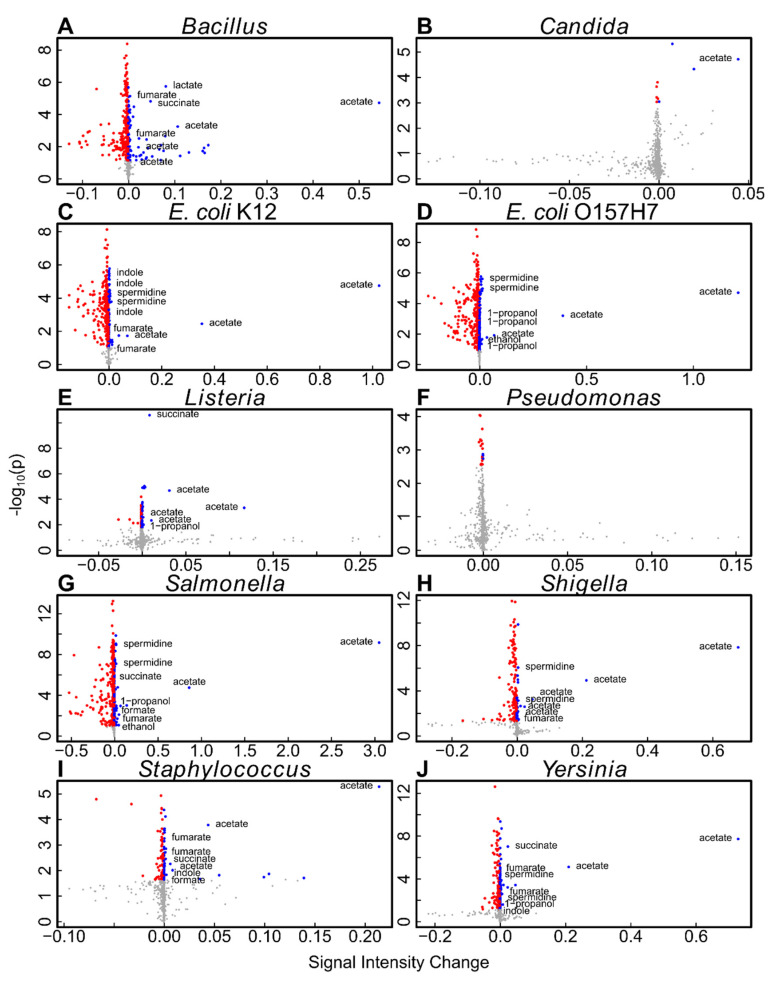
Volcano plots of strain-wise *t*-tests of the untargeted NMR data. Metabolites significant after FDR correction are colored in red (decreasing) or blue (increasing).

**Figure 6 metabolites-11-00863-f006:**
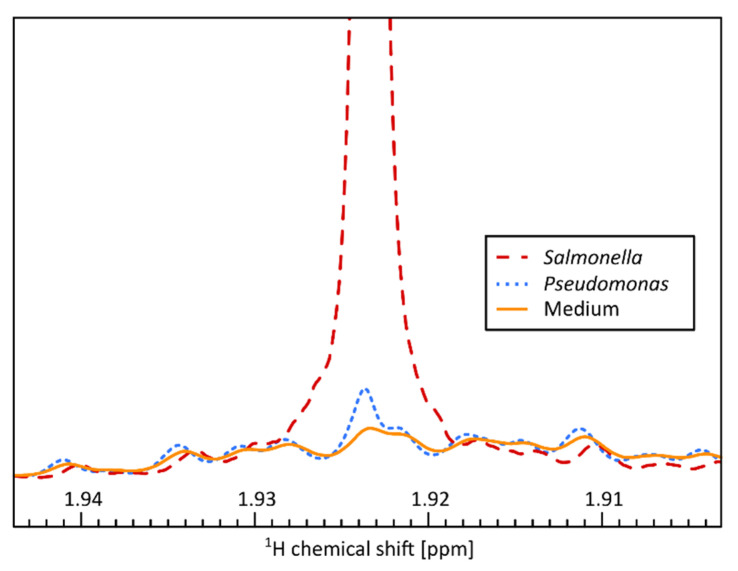
Acetic acid signal in the ^1^H NMR spectrum of *Salmonella* (red, dashed), *Pseudomonas* (blue, dotted), and medium (orange, solid).

**Table 1 metabolites-11-00863-t001:** Prediction accuracies of various methods, determined by outer cross-validation.

	Accuracy ^1^	
Method	Using All Signals	Using only Increasing Signals
Artificial neural networks (ANN)	91.2% ± 1.5%	99.2% ± 1.0%
Random forests (RF)	89.8% ± 2.2%	96.5% ± 1.6%
Support vector machines (SVM)	88.8% ± 0.0%	94.6% ± 0.0%

^1^ Mean ± standard deviation.

**Table 2 metabolites-11-00863-t002:** Prediction accuracies for artificial neural networks using only increasing signals.

		True									
		*Bacillus*	*Candida*	*E. coli*-K12	*E. coli*-O157H7	*Listeria*	*Pseudomonas*	*Salmonella*	*Shigella*	*Staphylococcus*	*Yersinia*
predicted	*Bacillus*	100%	0%	0%	0%	0%	0%	0%	0%	0%	0%
	*Candida*	0%	100%	0%	0%	0%	0%	0%	0%	0%	0%
	*E. coli*-K12	0%	0%	100%	0%	0%	0%	0%	3.3%	0%	0%
	*E. coli*-O157H7	0%	0%	0%	100%	0%	0%	0%	0%	0%	0%
	*Listeria*	0%	0%	0%	0%	95.2%	0%	0%	0%	0%	0%
	*Pseudomonas*	0%	0%	0%	0%	4.8%	100%	0%	0%	0%	0%
	*Salmonella*	0%	0%	0%	0%	0%	0%	100%	0%	0%	0%
	*Shigella*	0%	0%	0%	0%	0%	0%	0%	96.7%	0%	0%
	*Staphylococcus*	0%	0%	0%	0%	0%	0%	0%	0%	100%	0%
	*Yersinia*	0%	0%	0%	0%	0%	0%	0%	0%	0%	100%

Green indicates optimal accuracies, yellow indicates imperfect accuracies.

**Table 3 metabolites-11-00863-t003:** Metabolites that significantly increased during microbial growth for different microbes.

	*Bacillus*	*Candida*	*E. coli*-K12	*E. coli*-O157H7	*Listeria*	*Pseudomonas*	*Salmonella*	*Shigella*	*Staphylococcus*	*Yersinia*
Acetic acid	+	+	+	+	+	0	+	+	+	+
Ethanol	0	0	0	+	0	0	+	0	0	0
Formic acid	0	0	0	0	0	0	+	0	+	0
Fumaric acid	+	0	+	0	0	0	+	+	+	+
Indole	0	0	+	0	0	0	0	0	+	+
Lactic acid	+	0	0	0	0	0	0	0	0	0
1-Propanol	0	0	0	+	+	0	+	0	0	+
Spermidine	0	0	+	+	0	0	+	+	0	+
Succinic acid	+	0	0	0	+	0	+	0	0	+

+ (yellow) indicates metabolites that consistently increased in a microbe; 0 (blue) indicates metabolites that did not consistently increase during growth of a microbe.

## Data Availability

The employed data are available at https://doi.org/10.5281/zenodo.5765900.

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
