# Peer review of "Deep Learning for Rapid Identification of Microbes Using Metabolomics Profiles"

_metabolites, 2021, doi:10.3390/metabo11120863_

Round 1

Reviewer 1 Report

This is a nice demonstrative study of ANN to identify the microbes using NMR-based metabolomics profiles. The authors used a set of NMR data of 10 different microbes to build to the ANN network and self-predict (i.e., analyze the different NMR profiles in the different microbes). The manuscript is well written with a clear systematic description and approach to ANN. This is indeed beneficial would to the readers.

Few comments:

- as discussed in the discussion, ‘… application of this research could be the identification of microbial strains…’  I suppose it meant to identify the ‘unknown microbes’. This would be a very nice concept. Have you made this attempt? Indeed the quality control of the known samples (inputs) and the unknowns (outputs) are critical to making the prediction.

-aside from identification, could ANN also qualify the microbes? For example, take a known mixture of microbes with different ratios and use your ANN inputs built here to identify and quantify the microbes?  I suppose the quantification should be carried on a few targeted metabolite signals (i.e., table 2)?

Minor:

-line 128, why 2.25 times?  How did the authors obtain this value?

-line 133, similarly for 2x10-9?

Author Response

- as discussed in the discussion, ‘… application of this research could be the identification of microbial strains…’  I suppose it meant to identify the ‘unknown microbes’. This would be a very nice concept. Have you made this attempt? Indeed the quality control of the known samples (inputs) and the unknowns (outputs) are critical to making the prediction.

Reply: Yes, this is indeed what we intended to say and we have clarified this paragraph accordingly. We have not yet attempted to do so, as we expect some real-world issues that need to be solved first. We added a paragraph discussing these issues at the end of the discussion.

-aside from identification, could ANN also qualify the microbes? For example, take a known mixture of microbes with different ratios and use your ANN inputs built here to identify and quantify the microbes?  I suppose the quantification should be carried on a few targeted metabolite signals (i.e., table 2)?

Reply: This is a great suggestion and we have planned to attempt this in the future. One idea would be to first use ANN to identify microbes and then use general linear models to quantify microbes based on selected metabolites. We added this to the discussion.

Minor:

-line 128, why 2.25 times?  How did the authors obtain this value?

Reply: We chose this value empirically by analyzing the training behavior on a test data set. We added an explanation to the manuscript.

-line 133, similarly for 2x10-9?

Reply: Again, this was based on our test training results, and this value was in the range of the lowest non-zero values observed for loss in our data set. We added an explanation to the manuscript.

Reviewer 2 Report

The study is interesting, however the experiments mentioned in the manuscript do not involve real biological samples. In order to improve the manuscript, the authors can consider using this method for real biological problem.

Efforts to curb the spread of pathogens through food products include strict sanitary protocols, new technologies for inacativating food microbes, and testing for microbial contaminations.

Cite papers

Artificial neural networks (ANN) are a type of computational algorithm inspired by  the layered structure of the brain’s neurons and its ability to learn from, and classify, visual information.

Cite papers

Figure 2: Legend is short. Please elaborate

Initial tests showed that using two hidden layers led to higher accuracy as  compared to ANN with one hidden layer, while using 3 or more hidden layers did not 91 further increase accuracy

Mention accuracy number.

During training, the optimizer Adam (adaptive moment estimation) [7] yielded better accuracies than stochastic gradient descent (sgd)

Mention accuracy.

How are the metabolite levels normalized? How is the bacterial no taken into account in the NMR quantitation profile? Please mention that in text.

The authors have not tested the method in a real biological sample like stool. Can this method be used to know the bacterial constituents of stool? If so, please mention that in text.

Author Response

Efforts to curb the spread of pathogens through food products include strict sanitary protocols, new technologies for inacativating food microbes, and testing for microbial contaminations.

Cite papers

Reply: We added references for this section.

Artificial neural networks (ANN) are a type of computational algorithm inspired by  the layered structure of the brain’s neurons and its ability to learn from, and classify, visual information.

Cite papers

Reply: We added references for this section.

Figure 2: Legend is short. Please elaborate

Reply: Thank you for this remark, we have extended the figure legend.

Initial tests showed that using two hidden layers led to higher accuracy as  compared to ANN with one hidden layer, while using 3 or more hidden layers did not 91 further increase accuracy

Mention accuracy number.

Reply: We have added the respective accuracy increase.

During training, the optimizer Adam (adaptive moment estimation) [7] yielded better accuracies than stochastic gradient descent (sgd)

Mention accuracy.

Reply: We have added the respective accuracy increase.

How are the metabolite levels normalized? How is the bacterial no taken into account in the NMR quantitation profile? Please mention that in text.

Reply: We added these details to the Materials and Methods section.

The authors have not tested the method in a real biological sample like stool. Can this method be used to know the bacterial constituents of stool? If so, please mention that in text.

Reply: This is a really interesting idea! We believe that, in the future, this method will be able to do so, but the current models are not optimized for this application. We have added a discussion on future applications on real-life samples to the Discussion section.